# CTGC: CLUSTER-AWARE TRANSFORMER FOR GRAPH CLUSTERING

## ABSTRACT

Graph clustering is a fundamental unsupervised task in graph mining. However, mainstream clustering methods are built on graph neural networks, thus inevitably suffer from the difficulty in long-range dependencies capturing. Moreover, current two-stage clustering scheme, consisting of representation learning and clustering, limits the ability of the graph encoder to fully exploit task-related information, resulting in suboptimal embeddings. In this work, we propose CTGC (**C**luster-Aware **T**ransformer for **G**raph **C**lustering) to mitigate these issues. Specifically, considering the excellence of transformer in long-range dependencies modeling, we first introduce transformer to graph clustering as the crucial graph encoder. To further enhance the task awareness of encoder during representation learning, we presents two mechanisms: momentum cluster-aware attention and cluster-aware regularization. In momentum cluster-aware attention, previous clustering results are adopted to guide the node embedding production with specially designed cluster-aware queries. Cluster-aware regularization is designed to fuse the cluster information into bordering nodes through minimizing the overlap between different clusters while maximizing the completeness of each cluster. We evaluate our method on seven real-world graph datasets and achieve superior results compared to existing state-of-the-art methods, demonstrating its effectiveness in improving the quality of graph clustering.

## 1 INTRODUCTION

Graph clustering is an unsupervised task that partitions nodes into several distinct and non-overlapping clusters. It has numerous applications across various domains, including social networks and recommender systems. Currently, Graph Neural Networks (GNNs) methods, in conjunction with contrastive learning, are extensively employed for graph clustering. MVGRL (Hassani & Khasahmadi, 2020) uses graph diffusion to generate additional structural views and contrast them with regular views to learn node representations. BGRL (Thakoor et al., 2021) removes the requirement for negative sampling by minimizing an invariance-based loss on augmented graphs within each batch. Dink-Net (Liu et al., 2023b) initially pretrains the model by contrasting dropped and shuffled views, followed by fine-tuning that minimizes distances between samples and cluster centers, thereby drawing samples closer to the centers. MAGI (Liu et al., 2024) proposes to use modularity maximization as a contrastive pretext task to avoid the problem of semantic drift.

As indicated by the "no free lunch" theorem (Wolpert & Macready, 1997), the GNN-based encoders used in existing clustering methods exhibit significant limitations. Most GNNs are designed to be equivalent to first-order Weisfeiler-Lehman test, learning node representations by locally aggregating features of neighboring nodes in each layer, which makes it difficult to effectively capture long-range dependencies (Dai et al., 2018). While layer stacking has the potential to enhance long-range information propagation, it also introduces challenges such as over-smoothing (Chen et al., 2020a) and over-squashing (Alon & Yahav, 2021). To intuitively illustrate this issue, we selected three commonly used graph datasets (i.e., Cora, CiteSeer and PubMed) to analyze the shortest path distances between nodes within the same cluster. The results, presented in Figure 1, shows that a significant portion of the shortest path distances between nodes exceeds three, even within the same cluster. Most current graph clustering methods are designed with a model depth of only two or three layers, limiting their ability to fully propagate information. This underscores the need to account for long-range dependencies in clustering models.

Furthermore, current graph clustering methods predominantly adopt a two-stage clustering scheme. Typically, this involves using a GNN to encode raw node features into embeddings, followed by clustering using traditional methods like KMeans or spectral clustering to generate cluster assignments for evaluation. However, a critical flaw exists in this sheme. The representation learning and clustering stages are entirely decoupled within the framework, preventing the model from accessing sufficient task-specific information, namely, feedback on clustering results to produce more ef-

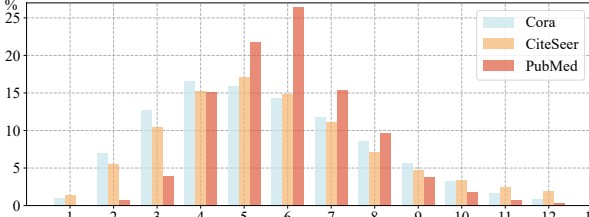

Figure 1: Data statistics of the shortest path distances between nodes within the same cluster on the Cora, CiteSeer and PubMed datasets. For better visualization, we truncated the part with $x \geq 13$, which is reasonable as they account for less than 2% of the total dataset.

fective embeddings. While several works try to alleviate this issue, they are primarily limited to utilizing clustering results from a regularization perspective (Liu et al., 2023b; 2024), with no integration of task-specific information in the forward pass of the model. Therefore, we decide to directly incorporate clustering information into the core mechanism of the model, specifically, the attention mechanism in our approach, to enhance the model's task awareness.

In this work, we propose CTGC (**C**luster-Aware **T**ransformer for **G**raph **C**lustering) to solve these issues. Specifically, we first introduce transformer in light of its superior ability of modeling long-range dependencies. To address the lack of task-related information during the representation learning stage, we propose momentum cluster-aware attention and cluster-aware regularization. Momentum cluster-aware attention uses prior clustering results to generate a cluster index for each node, then produces embeddings based on cluster-related queries, and finally assigns embeddings according to each node's cluster index. Furthermore, considering that there exists data points may be difficult to distinguish between multiple clusters, we propose cluster-aware regularization, which minimizes the overlap between different clusters while maximizing the completeness of each cluster. This enhancement of task-related information helps guide the model towards producing more coherent and accurate clusters. Overall, the main contributions are summarized as follows:

- In this work, we propose CTGC, to address the issues of long-range dependency and task-related information missing in current graph clustering methods.
- To the best of our knowledge, we are the first to introduce a pure attention-based transformer for graph clustering to alleviate the long-range dependency modeling.
- We propose two mechanism: momentum cluster-aware attention and cluster-aware regularization, to mitigate the issue of task-related information missing.
- Comprehensive experiments on seven real-world graph datasets are conducted to validate the effectiveness of our method in graph clustering.

## 2 RELATED WORK

**Graph Clustering.** Graph clustering is a widely studied problem in academia and industry. In recent years, contrastive learning has emerged as a prominent approach in graph clustering, with notable examples including MVGRL (Hassani & Khasahmadi, 2020), Dink-Net (Liu et al., 2023b), and MAGI (Liu et al., 2024). MVGRL leverages graph diffusion to generate alternative structural views and contrasts them with standard views to learn node representations. Dink-Net pretrains the model by contrasting dropped and shuffled views, followed by fine-tuning, where it minimizes the distances between samples and cluster centers, pulling samples closer to the respective centers. MAGI introduces modularity maximization as a contrastive pretext task to mitigate the issue of semantic drift. However, all of these methods follow a two-stage clustering scheme. This separation between representation learning and clustering causes the model to lose task-related information during embedding generation, ultimately limiting its performance. In our work, we integrate previous clustering results directly into the attention mechanism, and then propose two mechanisms (momentum cluster-aware attention and cluster-aware regularization) to alleviate this problem. More related work on graph clustering is discussed in Appendix A.1.

**Transformer in Graph.** Transformer (Vaswani et al., 2017) has achieved remarkable success in many fields such as computer vision and speech recognition. Recently, transformers emerge as an alternative technique for graph learning. So far, a great variety of transformers have been proposed to adapt graph structured data (Rong et al., 2020; Ying et al., 2021; Zhao et al., 2021; Xu et al., 2021; Chen et al., 2021; Wu et al., 2022; Chen et al., 2023; Liu et al., 2023a; Shomer et al., 2024; Rampášek et al., 2022; Nguyen et al., 2022). GROVER adopts a dynamic message passing strategy and randomly selects propagation hops at each layer. Gophormer samples ego-graphs and converts them into sequences as input to alleviate scalability issues. NodeFormer designs a kernelized Gumbel-Softmax operator to reduce the algorithm complexity w.r.t node numbers. NAGphormer proposes a novel neighborhood aggregation module to adaptively learn neighborhoods with different hops. Gapformer proposes to combine the attention mechanism with graph coarsening and only use pooled nodes to calculate attention. However, there is still none for graph clustering, and the excellence of transformer in long-range dependency modeling inspires us the solution for graph clustering. More related work on transformer in graph is provided in Appendix A.1.

## 3 NOTATIONS AND PRELIMINARIES

**Notations.** Consider a graph $G = (V, E)$ with vertex set $V = \{v_1, ..., v_n\}$, where $|V| = N$, and edge set $E \subseteq V \times V$, where $|E| = m$. Let $A \in \mathbb{R}^{n \times n}$ be the adjacency matrix of $G$, where $A_{ij} = 1$ if $(v_i, v_j) \in E$, and $A_{ij} = 0$ otherwise. Let $X \in \mathbb{R}^{n \times d}$ be the feature matrix, where the $i$-th row $X_i$ denotes the $d$-dimensional feature vector of node $i$.

**Graph Clustering.** Given the graph $G$ and node attributes $X$, the goal is to partition the graph $G$ into $|\mathbb{C}|$ partitions $\{\mathcal{C}_1, ..., \}$ such that nodes in the same cluster are similar/close to each other in graph structure and features. The current mainstream methods often use GNNs as the encoder, then optimize the problem and generate node embeddings under the framework of contrastive learning, and finally use traditional algorithms such as KMeans to generate cluster assignments for evaluation. Let $z_u$ and $z_{u^+}$ denote embeddings of a positive pair by a GNN encoder, we can then apply a loss function such as InfoNCE for contrastive learning, defined as follows:

$$\mathcal{L}_{\text{InfoNCE}} = -\frac{1}{N} \sum_{u=1}^{N} \log \frac{\exp(\text{sim}(z_u, z_{u^+})/\tau)}{\exp(\text{sim}(z_u, z_{u^+})/\tau) + \sum_{i=1}^{N_{neg}} \exp(\text{sim}(z_u, z_i)/\tau)} \quad (1)$$

where $\text{sim}(\cdot, \cdot)$ denotes the similarity function (often cosine similarity), $\tau$ is the adjustable temperature parameter that controls local separation and global uniformity and $N_{neg}$ is the number of negative samples. Ultimately, the clustering partition is obtained through $\mathcal{C} = f_C(Z)$, where $f_C(Z)$ represents a clustering method, such as KMeans or spectral clustering. To maintain simplicity in our framework, we only replace the GNN encoder with our momentum cluster-aware transformer and introduce a cluster-aware regularization.

## 4 METHODS

### 4.1 OVERVIEW OF CTGC

As illustrated in Figure 2, the core idea of CTGC is to capture long-range dependencies by replacing the basic graph encoder and to enhance task-related information by explicitly incorporating clustering information into it. Our method takes three steps: modeling long-range dependency, momentum cluster-aware attention and cluster-aware regularization. In the first step (§ 4.2), we introduce transformer to model long-range node dependencies. In the second step (§ 4.3), we first generate cluster embeddings using specially designed cluster-related queries, then assign them based on previous clustering results, and finally fuse them with standard attention. In the final step (§ 4.3), we introduce a regularization to handle cluster overlap. The underlying idea behind these improvements is similar to leveraging global information. In graph clustering, cluster information not only captures the overall structure information of a graph's substructure but also carries task-specific information, such as cluster assignments, cluster centers, and node embeddings. Effective utilization of cluster information helps guide the encoder to produce more suitable embeddings for clustering.

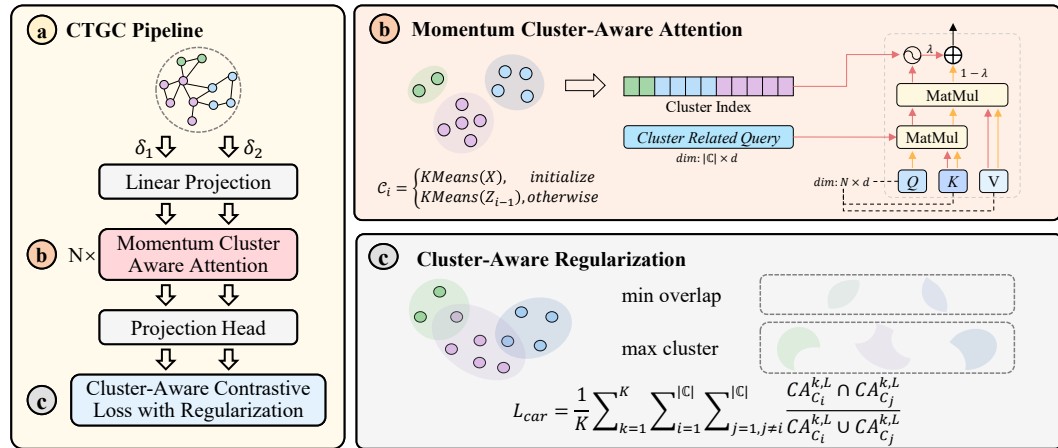

Figure 2: Overview of our proposed CTGC framework. The entire figure can be divided into three parts: (a), (b), and (c). Part (a) illustrates the overall pipeline, while parts (b) and (c) detail the improved modules we introduce. Briefly, we first apply dropout before linear projection to generate different initial features ($\delta_i$ denotes different dropping rates), then employ the transformer encoder based on momentum cluster-aware attention to produce diverse contrastive views, and finally conduct contrastive learning using both the base loss and the cluster-aware regularization.

## 4.2 Modeling Long-Range Dependencies

In this work, we employ transformer as the graph encoder based on its superior ability of modeling long-range dependencies. Most transformers are based on a multi-head self-attention module followed by a residual connection with a normalization layer. Let $d$ and $K$ denote the dimension of the feature space and the number of attention heads respectively. Formally, the standard self-attention uses three different matrices $W_Q \in \mathbb{R}^{d \times d_K}$, $W_K \in \mathbb{R}^{d \times d_K}$ and $W_V \in \mathbb{R}^{d \times d_K}$ to project input node features $X$ into corresponding representations of the query ($Q$), the key ($K$) and the value ($V$). The node embedding learning in transformer is described as follows:

$$z_u^{(l+1)} = \sum_{v=1}^{N} \tilde{a}_{uv}^{(l)} \cdot (W_V^{(l)} z_v^{(l)}), \quad \tilde{a}_{uv}^{(l)} = \frac{\exp((W_Q^{(l)} z_u^{(l)})^\top (W_K^{(l)} z_v^{(l)}))}{\sum_{w=1}^{N} \exp((W_Q^{(l)} z_u^{(l)})^\top (W_K^{(l)} z_w^{(l)}))} \tag{2}$$

where $W_Q^{(l)}$, $W_K^{(l)}$ and $W_V^{(l)}$ are different learnable parameters in the $l$-th layer. We omit the scaling factor $\sqrt{d_K}$ and the nonlinear activation after aggregation for brevity. Unlike GNNs, which propagate information through local neighborhood aggregation, the transformer's attention mechanism enables each node to interact directly with all other nodes, not just its neighbors. This allows the transformer to expand its receptive field to the entire graph with only a single layer, thereby capturing long-range dependencies more effectively.

## 4.3 Task-Related Information

We mainly present two mechanisms to enhance the missing task-related information in the representation learning stage of the graph clustering, namely momentum cluster-aware attention and cluster-aware regularization, which are introduced as below.

**Momentum Cluster-Aware Attention.** Current graph clustering methods typically follow a two-stage scheme, where the separation of clustering and representation learning restricts the model to acquire sufficient task-related information, thereby limiting to produce more effective embeddings. An intuitive idea is to incorporate the clustering results into the encoder forward computation. Inspired by the momentum update, we integrate the previous clustering results into the attention mechanism and propose momentum cluster-aware attention. In the initial phase, when there are no embedding outputs, the original node features are used to generate clustering assignments. The overall

definition is as follows:

$$\mathcal{C}_i = \begin{cases} KMeans(X), & initialize \\ KMeans(Z_{i-1}), otherwise \end{cases} \tag{3}$$

where $Z_{i-1}$ is the node embeddings by the encoder at the $(i\text{-}1)$-th epoch. In order to generate cluster embeddings in a simple yet effective way, we design a cluster-related query $Q_\mathcal{C}$, where $W_{Q_\mathcal{C}}^{(l)}$ is a learnable query in $\mathbb{R}^{|\mathbb{C}| \times d}$ and is initialized by sampling from $\mathcal{N}(0, 1)$. $W_K^{(l)}$ and $W_V^{(l)}$ are learnable parameters in the $l$-th layer, shared by momentum cluster-aware attention and standard attention. Then we can use $Q_\mathcal{C}$ to get the cluster-aware attention map as follows:

$$CA^{(l)} = \frac{\exp((W_{Q_\mathcal{C}}^{(l)} Z^{(l)})^\top (W_K^{(l)} Z^{(l)}))}{\exp((W_{Q_\mathcal{C}}^{(l)} Z^{(l)})^\top (W_K^{(l)} Z^{(l)}))}, \quad \tilde{ca}_{uv}^{(l)} = \frac{\exp(W_{Q_\mathcal{C}}^{(l)} z_u^{(l)})^\top (W_K^{(l)} z_v^{(l)}))}{\sum_{w=1}^N \exp((W_{Q_\mathcal{C}}^{(l)} z_u^{(l)})^\top (W_K^{(l)} z_w^{(l)}))} \tag{4}$$

where $\tilde{ca}_{uv}^{(l)}$ is the attention weight of node $u$ to node $v$ in the momentum cluster-aware attention map of layer $l$. Then we first assign the corresponding clustering embedding to each node according to the clustering result obtained by Equation 3, and finally fuse the cluster-aware attention embedding and normal attention embedding, which is defined as follows:

$$\mathbf{z}_u^{(l+1)} = (1 - \lambda) \sum_{v=1}^N \tilde{a}_{uv}^{(l)} \cdot (W_V^{(l)} \mathbf{z}_v^{(l)}) + \lambda I(\sum_{v=1}^N \tilde{ca}_{uv}^{(l)} \cdot (W_V^{(l)} \mathbf{z}_v^{(l)})) \tag{5}$$

where $\lambda$ is the weight of cluster-aware attention embedding, and $I(\cdot)$ is the function that assigns the corresponding clustering embedding to each node according its clustering index. By tuning the value of $\lambda$, we can adjust the model's utilization of clustering information. Clustering information is similar to global information, so this fusion is like a trade-off between local and global information.

**Cluster-Aware Regularization.** As shown in Figure 2, there exists nodes may be difficult to distinguish between multiple clusters. Without additional constraints on these nodes, they may contribute to the generation of embeddings for multiple clusters, negatively impacting the model's final performance. A straightforward solution is to minimize the overlap between the cluster-aware attention maps of different clusters, as defined in Equation 4. In CTGC, we utilize the output from the final layer of the model. Assuming the model has a depth of $L$, the cluster overlap is defined as follows:

$$Overlap_{\mathcal{C}_i, \mathcal{C}_j} = CA_{\mathcal{C}_i}^L \cap CA_{\mathcal{C}_j}^L \tag{6}$$

The utilization of minimizing overlap can effectively reduce fuzzy boundary nodes, but simply using it may come with a side effect of reducing the area of each cluster. Considering the coverage area, the following properties typically hold true:

$$CA_{\mathcal{C}_i}^L + CA_{\mathcal{C}_j}^L = CA_{\mathcal{C}_i}^L \cup CA_{\mathcal{C}_j}^L - CA_{\mathcal{C}_i}^L \cap CA_{\mathcal{C}_j}^L \tag{7}$$

We can simultaneously maximize $CA_{\mathcal{C}_i}^L \cup CA_{\mathcal{C}_j}^L$ while minimizing the overlap, and the final cluster-aware regularization is defined as Equation 8.

$$L_{car} = \frac{1}{K} \sum_{k=1}^K \sum_{i=1}^{|\mathbb{C}|} \sum_{j=1, j \neq i}^{|\mathbb{C}|} \frac{CA_{\mathcal{C}_i}^{k,L} \cap CA_{\mathcal{C}_j}^{k,L}}{CA_{\mathcal{C}_i}^{k,L} \cup CA_{\mathcal{C}_j}^{k,L}} \tag{8}$$

where $K$ is the number of heads for momentum clustering-aware attention and standard attention.

### 4.4 Learning Objective

To keep the architecture simple, we just use the transformer encoder to replace the GNN, and the overall framework is similar to SimCLR (Chen et al., 2020b). To align with common experimental practices, we employ cosine similarity to assess the similarity between different embeddings, denoted as $\text{sim}(u, v) = u^T v / \|u\| \|v\|$, and subsequently utilize the InfoNCE loss, as defined in Equation 1, as the base loss function. The final optimization goal is a weighted sum of base loss and cluster-aware regularization, which is defined as follows:

$$L = (1 - \alpha) L_{base} + \alpha L_{car} \tag{9}$$

where $\alpha$ is the weight of cluster-aware attention regularization. By adjusting the value of $\alpha$, we can control the model's emphasis on the overlap between cluster nodes.

## 5 EXPERIMENTS

### 5.1 EXPERIMENT SETUP

**Environment and Datasets.** We use a single NVIDIA A100 GPU (40GB) and the PyTorch platform. Detailed model settings and hyperparameter values can be found in Appendix A.2. We assess our method on seven real world datasets (Kipf & Welling, 2017; Shchur et al., 2018), the details are presented in Table 1.

Table 1: Dataset statistics.

| Dataset | Nodes | Edges | Features | Clusters |
|---------|-------|-------|----------|----------|
| Cora | 2,708 | 5,278 | 1,433 | 7 |
| CiteSeer | 3,327 | 4,552 | 3,703 | 6 |
| PubMed | 19,717 | 44,324 | 500 | 3 |
| Amazon-Photo | 7,650 | 119,081 | 745 | 8 |
| Amazon-Computers | 13,752 | 245,861 | 767 | 10 |
| Coauthor-CS | 18,333 | 81,894 | 6,805 | 15 |
| Coauthor-Physics | 34,493 | 247,962 | 8,415 | 5 |

**Baseline.** We compare our method with eleven baselines, which can be categorized into two groups: **(1) Structure/features only methods**: Node2vec (Grover & Leskovec, 2016) and KMeans (MacQueen et al., 1967). **(2) Contrastive learning methods**: GRACE (Zhu et al., 2020), MVGRL (Hassani & Khasahmadi, 2020), BRGL (Thakoor et al., 2021), Dink-Net (Liu et al., 2023b), S³GC (Devvrit et al., 2022), CCGC (Yang et al., 2023), SCGDN (Ma & Zhan, 2023), DGCLUSTER (Bhowmick et al., 2024) and MAGI (Liu et al., 2024).

**Metrics.** We follow the evaluation setup of MAGI (Liu et al., 2024) and measure four metrics related to evaluating the quality of cluster assignments: Accuracy (ACC), Normalized Mutual Information (NMI), Adjusted Rand Index (ARI) and Macro-F1 Score (F1). For all the metrics, higher values indicate better performance. In our experiments, we first generate representations for each method and then perform KMeans clustering on the dataset to produce cluster assignments for evaluation.

### 5.2 EXPERIMENTAL RESULTS

**Graph Clustering Results.** Table 2 compares the clustering performance of CTGC with eleven strong baseline methods on seven real-world graph datasets. The results of baselines are mainly derived from (Liu et al., 2023b; 2024), except for three datasets (PubMed, Coauthor-CS, and Coauthor-Physics), whose results are reproduced based on the official implementation. For small-scale datasets, i.e., Cora, CiteSeer, we observe that MVGRL and Dink-Net are the two most competitive baseline methods. Nevertheless, CTGC outperforms them in all cases. For remaining datasets, CTGC significantly outperforms recent state-of-the-art baseline methods such as CCGC, DGCLUSTER, and MAGI. Compared to the runner-up, CTGC is ∼2.48% better on Cora, ∼1.63% better on CiteSeer, ∼1.10% better on Amazon-Photo, ∼6.99% better on Amazon-Computers, ∼3.88% better on Coauthor-CS and ∼4.93% better on Coauthor-physics in terms of clustering ACC. It is worth noting that CTGC has a huge improvement over the runner-up in the Coauthor-Physics dataset, with ACC increased by about 4.93%, NMI increased by about 4.69%, ARI increased by about 5.03%, and F1 score increased by about 6.20%. One possible explanation for the performance improvement is the high graph density of the Coauthor-Physics dataset tends to cause cluster overlap, while our proposed cluster-aware regularization can effectively alleviate this problem, and thus generate embeddings that are more suitable for clustering.

**t-SNE Visualization.** We use t-SNE to measure the quality of the generated embeddings. The embeddings generated by each method are projected into two-dimensional vectors for visualization. We select six strong baseline methods and raw features for visualization analysis. The visualization in Figure 3 intuitively shows that CTGC not only generates better cluster embeddings than the baseline methods, but also effectively discovers potential substructures in clusters.

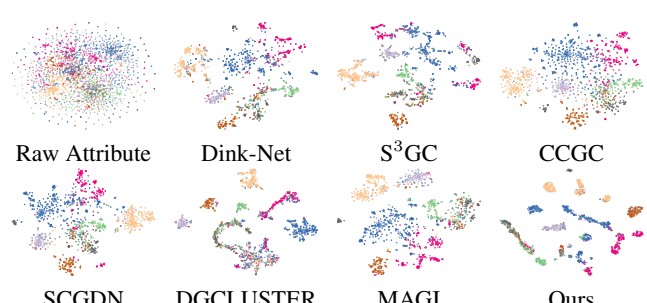

Figure 3: t-SNE visualization of CTGC along with six strong baselines and raw features on the Cora dataset.

Table 2: Clustering performance of CTGC and eleven baselines. The **bold** and underlined scores indicate the best and second best results respectively. "OOM" denotes out of memory.

| Dataset | Metric | Baselines | | | | | | | | | | | Ours |
| --- | --- | --- | --- | --- | --- | --- | --- | --- | --- | --- | --- | --- | --- |
| | | KMeans | Node2vec | GRACE | MVGRL | BGRL | Dink-Net | S³GC | CCGC | SCGDN | DGCLUSTER | MAGI | CTGC |
| Cora | ACC | 35.00 | 61.20 | 73.90 | 76.30 | 74.20 | 78.10 | 74.20 | 73.73 | 74.80 | 75.30 | 76.00 | **80.58** |
| | NMI | 17.30 | 44.40 | 57.00 | 60.80 | 58.40 | 62.28 | 58.80 | 55.93 | 56.90 | 60.00 | 59.70 | **62.37** |
| | ARI | 12.70 | 32.90 | 52.70 | 56.60 | 53.40 | 61.61 | 54.40 | 51.52 | 52.60 | 54.80 | 57.30 | **63.78** |
| | F1 | 36.00 | 62.10 | 72.50 | 71.60 | 69.10 | 72.66 | 72.10 | 72.70 | 70.40 | 70.60 | 73.90 | **78.37** |
| CiteSeer | ACC | 39.32 | 42.10 | 63.10 | 70.30 | 67.50 | 70.36 | 68.80 | 69.61 | 69.60 | 70.93 | 70.60 | **72.56** |
| | NMI | 19.90 | 24.00 | 39.90 | 45.90 | 42.20 | 45.87 | 44.10 | 44.12 | 44.30 | 45.36 | 45.20 | **46.63** |
| | ARI | 14.20 | 11.60 | 37.70 | 47.10 | 42.80 | 46.96 | 44.80 | 44.03 | 45.40 | 46.36 | 46.80 | **48.78** |
| | F1 | 39.40 | 40.10 | 60.30 | 65.40 | 63.10 | 65.96 | 64.30 | 62.70 | 65.50 | 65.13 | 64.80 | **66.63** |
| PubMed | ACC | 60.10 | 64.10 | 63.70 | 67.50 | 65.40 | 69.31 | 71.30 | 67.43 | 68.25 | 78.27 | 68.81 | **78.36** |
| | NMI | 31.40 | 28.80 | 30.80 | 34.50 | 31.50 | 28.14 | 34.56 | 30.98 | 28.56 | 38.31 | 32.92 | **42.55** |
| | ARI | 28.10 | 25.80 | 27.60 | 31.00 | 28.50 | 29.77 | 36.27 | 29.56 | 29.33 | 45.73 | 31.60 | **45.91** |
| | F1 | 59.20 | 63.40 | 62.80 | 67.20 | 64.90 | 67.84 | 70.42 | 67.27 | 66.90 | 77.21 | 68.46 | **78.24** |
| Photo | ACC | 27.22 | 27.58 | 67.66 | 50.91 | 66.54 | 81.71 | 75.20 | 77.53 | 78.00 | 82.00 | 79.00 | **83.10** |
| | NMI | 13.23 | 11.53 | 53.46 | 43.22 | 60.11 | 74.36 | 71.60 | 66.68 | 69.40 | 73.50 | 71.60 | **74.61** |
| | ARI | 5.50 | 4.92 | 42.74 | 28.62 | 44.14 | 68.40 | 56.10 | 58.96 | 60.70 | 67.10 | 61.50 | **70.76** |
| | F1 | 23.96 | 21.52 | 60.30 | 43.71 | 63.08 | 73.92 | 72.90 | 71.59 | 71.60 | 75.20 | 72.90 | **78.84** |
| Computer | ACC | 22.50 | 35.60 | 51.90 | 41.64 | 46.90 | 53.02 | 58.80 | 59.73 | 58.20 | 58.26 | 62.00 | **68.99** |
| | NMI | 11.00 | 27.80 | 53.80 | 35.06 | 44.10 | 32.95 | 56.00 | 54.64 | 54.50 | 52.03 | 59.20 | **59.32** |
| | ARI | 5.60 | 24.80 | 34.30 | 27.77 | 30.60 | 34.43 | 43.80 | 41.15 | 43.00 | 42.69 | 46.20 | **52.05** |
| | F1 | 15.20 | 22.40 | 39.00 | 33.00 | 41.50 | 39.45 | 47.50 | 50.45 | 48.00 | 48.42 | 57.40 | **59.01** |
| CS | ACC | 56.54 | 60.71 | 75.45 | 66.11 | 71.67 | 70.59 | 73.77 | 71.71 | 71.71 | 84.21 | 81.99 | **88.09** |
| | NMI | 57.88 | 62.08 | 74.34 | 65.32 | 72.03 | 61.51 | 73.60 | 75.78 | 74.13 | 78.22 | 78.68 | **83.06** |
| | ARI | 38.00 | 48.41 | 72.12 | 68.14 | 70.05 | 59.99 | 71.63 | 64.41 | 73.09 | 82.18 | 78.43 | **83.56** |
| | F1 | 41.20 | 58.13 | 69.66 | 62.29 | 65.55 | 63.32 | 64.02 | 68.83 | 60.22 | 68.85 | 74.35 | **81.48** |
| Physics | ACC | 44.07 | 58.48 | 87.75 | 78.56 | 82.95 | 84.15 | 77.06 | 88.23 | OOM | 81.94 | 87.25 | **93.16** |
| | NMI | 37.63 | 54.85 | 73.23 | 61.01 | 69.18 | 58.40 | 64.10 | 74.40 | | 72.55 | 70.45 | **79.09** |
| | ARI | 14.00 | 41.42 | 79.60 | 71.00 | 75.19 | 67.52 | 54.59 | 81.18 | | 80.46 | 78.14 | **86.21** |
| | F1 | 44.43 | 56.98 | 83.11 | 62.68 | 78.51 | 77.41 | 78.16 | 84.96 | | 80.43 | 82.25 | **91.16** |

## 5.3 ABLATION STUDY

**Ablation Experiment of Proposed Modules.** We conduct ablation studies to explore the efficacy of different components proposed by CTGC. We set two variants of the model for comparison and results are shown in Table 3. In Table 3, we observe that each improvement of the model has an impact on the final performance. When momentum cluster-aware attention is removed, the ACC of CTGC decreases by ∼1.41% on Cora, ∼1.99% on the PubMed, ∼3.41% on the Amazon-Computers, ∼3.14% on the Coauthor-CS and ∼1.02% on the Coauthor-Physics. The model performance drops drastically after removing momentum cluster-aware attention and cluster-aware regularization, for example, the F1 on Amazon-Computers and Coauthor-Physics dropped by ∼6.82% and ∼5.35%, respectively. This phenomenon also strongly verifies our motivation. As shown in Table 3, after removing all proposed improvements, that is, in the $V_3$ version, the performance of the transformer still remains superior to most GNN methods listed in Table 2, which further emphasizes the effectiveness of the long-range dependency modeling in graph clustering tasks. The introduction of additional task-related information during the representation learning stage results in a significant enhancement in Transformer performance, surpassing the previous state-of-the-art GNN methods.

Table 3: Ablation studies of proposed modules, where MCAA denotes momentum cluster-aware attention and CAR denotes cluster-aware regularization. The **bold** and underlined scores indicate the best and second best results respectively. $V_i$ denotes different versions of the model. Cluster-aware regularization depends on momentum cluster-aware attention, so in $V_3$, removing momentum cluster-aware attention also implicitly removes cluster-aware regularization.

| Datasets | Choices | ACC | NMI | ARI | F1 |
| --- | --- | --- | --- | --- | --- |
| Cora | $V_1$: CTGC | **80.58** | **62.37** | **63.78** | **78.37** |
| | $V_2$: w/o CAR | 79.17 | 61.29 | 60.31 | 76.76 |
| | $V_3$: w/o MCAA (& CAR) | 78.41 | 60.08 | 57.39 | 76.01 |
| CiteSeer | $V_1$: CTGC | **72.56** | **46.63** | **48.78** | **66.63** |
| | $V_2$: w/o CAR | 71.81 | 44.80 | 47.26 | 65.39 |
| | $V_3$: w/o MCAA (& CAR) | 71.23 | 43.80 | 47.06 | 64.94 |
| PubMed | $V_1$: CTGC | **78.36** | **42.55** | **45.91** | **78.24** |
| | $V_2$: w/o CAR | 76.37 | 40.07 | 43.61 | 76.52 |
| | $V_3$: w/o MCAA (& CAR) | 74.12 | 38.50 | 42.37 | 75.89 |
| Photo | $V_1$: CTGC | **83.10** | **74.61** | **70.76** | **78.84** |
| | $V_2$: w/o CAR | 82.20 | 72.96 | 69.49 | 77.52 |
| | $V_3$: w/o MCAA (& CAR) | 80.29 | 71.76 | 67.41 | 75.26 |
| Computer | $V_1$: CTGC | **68.99** | **59.32** | **52.05** | **59.01** |
| | $V_2$: w/o CAR | 65.58 | 56.19 | 48.82 | 57.05 |
| | $V_3$: w/o MCAA (& CAR) | 64.69 | 54.16 | 45.23 | 53.90 |
| CS | $V_1$: CTGC | **88.09** | **83.06** | **83.56** | **81.48** |
| | $V_2$: w/o CAR | 84.95 | 81.29 | 81.51 | 79.08 |
| | $V_3$: w/o MCAA (& CAR) | 82.40 | 80.18 | 80.64 | 74.84 |
| Physics | $V_1$: CTGC | **93.16** | **79.09** | **86.21** | **91.16** |
| | $V_2$: w/o CAR | 92.14 | 77.14 | 83.24 | 90.22 |
| | $V_3$: w/o MCAA (& CAR) | 88.53 | 74.42 | 80.86 | 87.98 |

**Comparison of Cluster Embedding Generation Methods.** We also compare our approach of generating cluster embeddings using momentum cluster-aware attention with three common and intuitive methods, and the results are shown in Table 4. Compared to the runner-up, our approach is ∼4.69% better on Cora, ∼8.98% better on PubMed, ∼6.26% better on Amazon-Photo, and ∼5.29% better on Coauthor-CS in terms of ACC. It is worth noting that on the CiteSeer and Coauthor-Physics datasets, the improvement of our method compared with Avg is not as significant as on other datasets. This is because in these two data sets, there are relatively few dependencies between different clusters, which is also reflected in the subsequent visualization of attention weights in Figure 4. As can be seen from Table 4, Sum performs poorly in most cases, which is consistent with our intuition that different clusters are likely to show similar results when summing the nodes within the cluster. Avg performs better than Max in most cases, probably because most of the nodes within the cluster are similar, so in this case Max's distinguishing ability is inferior than Avg. Compared with Avg, Max and Sum, our momentum cluster-aware attention only requires some additional cluster-related queries, which is also simple and plug-and-play.

**Attention Weight Visualization.** We further selecte three commonly used graph datasets (CiteSeer, PubMed and Coauthor-Physics) to visualize the attention weight between nodes for analysis. The visualization results are shown in Figure 5.3 and the the visualization results of the remaining graph datasets can be found in Appendix A.3. Comparing $V_1$ with $V_2$, we observe that the removal of cluster-aware regularization increases the inter-cluster dependencies in the lower right corner of the diagonal across all three datasets. This phenomenon also manifests between the first and second clusters in the upper left corner of the Coauthor-Physics dataset. This supports our claim and motivation that cluster-aware regularization effectively reduces overlap between different clusters. Further, when momentum cluster-aware attention is removed, as seen in the comparison between $V_1$ and $V_3$, the clusters at both ends of the diagonal become largely indistinguishable, underscoring the importance of our proposed modules. Comparing $V_2$ with $V_3$, we observe that after removing the momentum cluster-aware attention, the two clusters in the lower right corner of the diagonal of version V3 become more similar, indicating that in the absence of momentum cluster-aware attention, the model's ability to distinguish clusters is significantly weakened.

Table 4: Clustering performance of different cluster embeddings. The **bold** and underlined scores indicate the best and second best results.

| Datasets | Choices | ACC | NMI | ARI | F1 |
|---|---|---|---|---|---|
| Cora | Ours | **80.58** | **62.37** | **63.78** | **78.37** |
| | Avg | 75.89 | 58.18 | 55.92 | 73.65 |
| | Max | 75.81 | 57.36 | 56.82 | 73.34 |
| | Sum | 67.50 | 45.14 | 42.58 | 65.26 |
| CiteSeer | Ours | **72.29** | **46.28** | **48.78** | **66.38** |
| | Avg | 71.54 | 45.20 | 46.95 | 65.43 |
| | Max | 66.85 | 39.02 | 39.39 | 62.62 |
| | Sum | 66.76 | 38.75 | 38.81 | 60.89 |
| PubMed | Ours | **78.36** | **42.55** | **45.91** | **78.24** |
| | Avg | 69.38 | 36.44 | 32.32 | 69.23 |
| | Max | 62.96 | 28.02 | 25.51 | 64.10 |
| | Sum | 65.36 | 29.46 | 26.70 | 66.31 |
| Photo | Ours | **83.10** | **74.61** | **70.76** | **78.84** |
| | Avg | 75.56 | 66.45 | 56.64 | 71.92 |
| | Max | 76.84 | 64.81 | 56.72 | 74.56 |
| | Sum | 73.29 | 59.22 | 55.72 | 65.21 |
| Computer | Ours | **68.99** | **59.32** | **52.05** | **59.01** |
| | Avg | 50.77 | 43.05 | 37.90 | 39.15 |
| | Max | 44.61 | 38.73 | 33.13 | 33.05 |
| | Sum | 46.85 | 35.21 | 34.24 | 34.36 |
| CS | Ours | **88.09** | **83.06** | **83.56** | **81.48** |
| | Avg | 80.49 | 79.05 | 78.68 | 71.03 |
| | Max | 82.40 | 78.66 | 77.84 | 73.37 |
| | Sum | 73.78 | 73.58 | 73.61 | 61.81 |
| Physics | Ours | **93.13** | **79.09** | **86.21** | **91.16** |
| | Avg | 92.82 | 78.30 | 85.01 | 90.84 |
| | Max | 89.02 | 71.79 | 81.10 | 84.45 |
| | Sum | 81.37 | 61.18 | 68.13 | 77.49 |

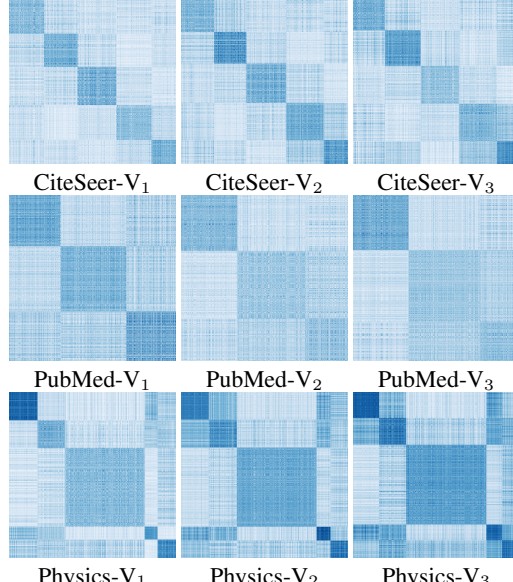

| CiteSeer-$V_1$ | CiteSeer-$V_2$ | CiteSeer-$V_3$ |
| PubMed-$V_1$ | PubMed-$V_2$ | PubMed-$V_3$ |
| Physics-$V_1$ | Physics-$V_2$ | Physics-$V_3$ |

Figure 4: Attention visualization on the CiteSeer, PubMed and Coauthor-Physics datasets. Results of the remaining graph datasets can be found in Appendix A.3. The color shade represents the different attention weight values. The darker the color, the greater the value. The clearly visible squares on the diagonal correspond to the clustering assignments generated by ours CTGC.

## 5.4 CASE STUDY

**Visualization of Masked and Unmasked Attention Weights.** To highlight the impact of long-distance dependencies captured by the transformer, we performed case studies on the CiteSeer, PubMed, and Coauthor-Physics datasets. Specifically, we reset the attention weights between nodes within the same cluster with a shortest path length greater than 3 to 0 (Masked) and compared these with the original attention weights (Unmasked). The results are presented in Figure 5.3. Comparing the Masked and Unmasked versions, it is evident that the cluster separation becomes less distinct after resetting the attention weights to 0, resulting in more blurred cluster boundaries. Additionally, in the Masked version, the influence between different clusters is amplified compared to the Unmasked version. This occurs because, in the Unmasked version, intra-cluster attention is reinforced, enhancing the cohesion within each cluster. When this reinforcement is removed, the corresponding attention weights are reduced, diminishing the differences between clusters and making them harder to distinguish. This fully demonstrates the benefits and importance of capturing long-range dependencies in clustering tasks.

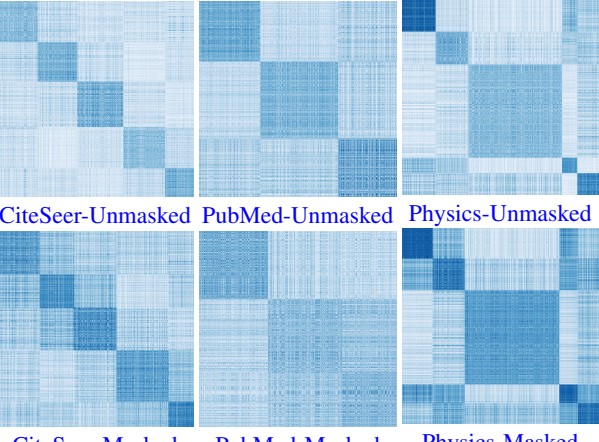

CiteSeer-Unmasked  PubMed-Unmasked  Physics-Unmasked

CiteSeer-Masked  PubMed-Masked  Physics-Masked

Figure 5: Visualization of Masked and Unmasked Attention Weights on the CiteSeer, PubMed and Coauthor-Physics datasets. "Unmasked" and "Masked" represent whether the attention weights for nodes with a shortest path length greater than 3 are reset to 0, aimed at evaluating the effectiveness of modeling long-range dependencies. The color shade represents the different attention weight values. The darker the color, the greater the value. The clearly visible squares on the diagonal correspond to the clustering assignments generated by ours CTGC.

## 5.5 SENSITIVITY ANALYSIS

We conduct extensive experiments to examine the sensitivity of hyperparameters of different components in CTGC. There are two main hyperparameters, the cluster-aware attention weight $\lambda$ and the cluster-aware regularization weight $\alpha$.

**Sensitivity Analysis of The Momentum Cluster-Aware Attention Weight $\lambda$.** We measure different results for $\lambda$ ranging from 0 and 0.9. Figure 6 shows our results. As can be seen, best results are obtained when $\lambda$ is about 0.1 or 0.2. We believe that the momentum cluster-aware attention is a kind of task-related global information, so the $\lambda$ cannot be too large, otherwise the embedding will become a representation of the cluster rather than the node, which is not suitable for clustering.

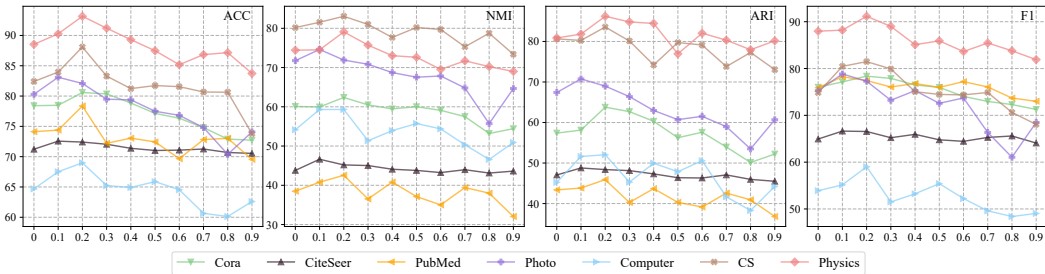

Figure 6: Sensitivity analysis of the momentum cluster-aware attention weight $\lambda$.

**Sensitivity Analysis of The Cluster-Aware Regularization Weight** $\alpha$**.** We measure different results for $\alpha$ ranging from 0 and 0.9. Figure 7 shows our results. As can be seen, best results are obtained when $\alpha$ is about 0.1 or 0.2. We think that the cases where nodes are indistinguishable between multiple clusters are only a small fraction of the total, so adding some constraints will have positive benefits, but when $\alpha$ is too large, it will make the model ignore learning node embeddings.

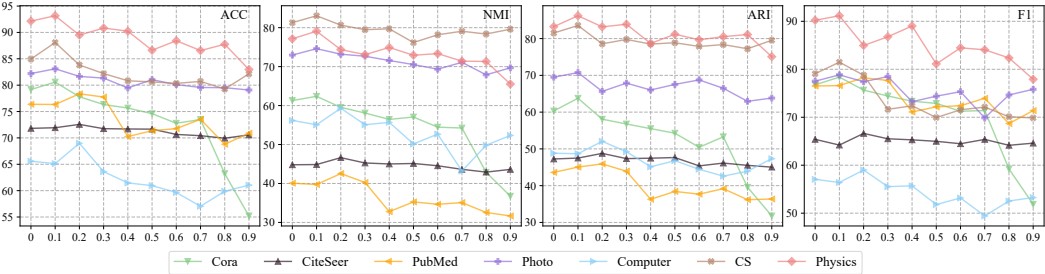

Figure 7: Sensitivity analysis of the cluster-aware regularization weight $\alpha$.

## 6  CONCLUSION

In this paper, we fully explore transformer for graph clustering. Mainstream clustering methods are built with GNNs, thus inevitably suffer from the difficulty in effectively long-range dependencies capturing. To address this, we introduce transformer to graph clustering in light of its ability of modeling long-range dependencies. Moreover, the prevailing two-stage clustering scheme, consisting of representation learning and nodes clustering, limits the graph encoder's capacity to fully utilize task-specific information, leading to suboptimal embeddings. Thus we propose momentum cluster-aware attention and cluster-aware regularization. Momentum cluster-aware attention utilizes previous clustering results to generate cluster indices for each node, produce embeddings based on cluster-related queries, and assign cluster-aware embeddings accordingly. Cluster-aware regularization minimizes the overlap between clusters while maximizing cluster completeness, ensuring that cluster information is correctly propagated to neighboring nodes. Extensive experiments on seven real-world graph datasets demonstrate the effectiveness of our method, which achieves state-of-the-art results compared to eleven strong baselines.

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

# A APPENDIX

## A.1 MORE RELATED WORK

**Graph Clustering.** Traditional clustering methods usually involve solving optimization problems or using some heuristic, non-parametric methods, such as KMeans (MacQueen et al., 1967), spectral clustering (Shi & Malik, 2000) and Louvain (Blondel Vincent et al., 2008). With the rise of deep learning, random walk-based methods such as DeepWalk (Perozzi et al., 2014) and Node2vec (Grover & Leskovec, 2016) have also been introduced for addressing clustering tasks. However, these methods usually only utilize features or structure, which limits their performance.

Thanks to the powerful expressiveness of GNNs, there have been many attempts to use GNNs for graph clustering. MGAE (Wang et al., 2017) applies autoencoders to graph learning and then proposes a marginalized GNN. DAEGC (Wang et al., 2019) proposes a unified goal-oriented framework to jointly optimize autoencoder embedding and clustering learning. SDCN (Bo et al., 2020) first constructs K-Nearest Neighbor graphs, which are then combined with the raw data and fed into the model. With the designed delivery operator, SDCN can effectively integrate structure-aware and autoencoder-specific representation.

In the past few years, contrastive learning has become a hotspot in graph clustering such as GRACE (Zhu et al., 2020), MVGRL (Hassani & Khasahmadi, 2020), BRGL (Thakoor et al., 2021), Dink-Net (Liu et al., 2023b), $S^3GC$ (Devvrit et al., 2022), CCGC (Yang et al., 2023), SCGDN (Ma & Zhan, 2023), DGCLUSTER (Bhowmick et al., 2024) and MAGI (Liu et al., 2024). GRACE maximizes the agreement of node representations between two corrupted views of a graph. $S^3GC$ uses a single GNN layer with the normalized adjacency and diffusion matrices, which can consider high-order neighborhood information. BGRL eliminates the need for negative sampling by minimizing an invariance-based loss for augmented graphs within a batch. MVGRL, Dink-Net and MAGI have already been stated clearly in the main text, so there is no repeation.

**Transformer in Graph.** Transformer (Vaswani et al., 2017) has achieved remarkable success in many fields such as computer vision and speech recognition. Recently, transformers emerge as an alternative technique for graph learning. So far, a great variety of transformers have been proposed to adapt to different levels of graph structured data.

For node-level tasks, Graphormer (Ying et al., 2021) proposes three structural encodings to embed graph structure information. Gophormer (Zhao et al., 2021) samples ego-graphs and converts them into sequences as input to alleviate scalability issues. NodeFormer (Wu et al., 2022) designs a kernelized Gumbel-Softmax operator to reduce the algorithm complexity w.r.t node numbers. NAG-phormer (Chen et al., 2023) proposes a novel neighborhood aggregation module to adaptively learn neighborhoods with different hops. Gapformer (Liu et al., 2023a) proposes to combine the attention mechanism with graph coarsening and only use pooled nodes to calculate attention.

For edge-level tasks, TRRN (Xu et al., 2021) proposes a relational reasoning network with dynamic memory based on the policy network enhanced by differentiable binary routers. HittER (Chen et al., 2021) proposes a hierarchical transformer model to jointly learn entity-relation combination and relation contextualization. LPFormer (Shomer et al., 2024) uses the attention mechanism to model all possible link factors and adaptively learns pairwise encodings between nodes by modeling multiple factors of the link prediction integral.

For graph-level tasks, GROVER (Rong et al., 2020) adopts a dynamic message passing strategy and randomly selects propagation hops at each layer. GraphGPS (Rampášek et al., 2022) proposes a linear modular framework by decoupling the local real edge aggregation and the transformer. UG-former (Nguyen et al., 2022) samples different neighbors in each batch and minimizes the sampled softmax loss, allowing the model to identify and distinguish structural differences. To our best knowledge, there are still none for graph clustering and we decide to make some attempts.

## A.2 NOTATION AND DETAILED EXPERIMENTAL SETTINGS

**Notations.** As an expansion of Section 5.1, we summarize the frequently used notations in Table 5.

Table 5: The most frequently used notations in this paper.

| Notation | Meaning |
|---|---|
| $G$ | Attribute Graph |
| $K$ | Attention Heads Number |
| $L$ | Model Depth |
| $d$ | Latent Feature Dimension Number |
| $N$ | Nodes Number |
| $\{\mathcal{C}_1, ..., \}$ | Cluster Assignment Matrices |
| $|\mathbb{C}|$ | Cluster Number |
| $A \in \mathbb{R}^{n \times n}$ | Adjacency Matrix |
| $X \in \mathbb{R}^{n \times d}$ | Attribute Matrix |
| $Q_{\mathcal{C}}$ | Cluster-Related Query |
| $W_Q, W_K, W_V \in \mathbb{R}^{d \times d_K}$ | Attention Matrix (Query/Key/Value) |
| $CA$ | Cluster-Aware Attention Map |
| $I(\cdot)$ | A Cluster Embedding Assignment Function |
| $f_{\mathcal{C}}(\cdot)$ | Traditional Clustering Methods (i.e., KMeans) |
| $\delta$ | The Dropping Rate |
| $\lambda$ | The Momentum Cluster-Aware Attention Weight |
| $\alpha$ | The Cluster-Aware Regularization Weight |
| $z$ | A Node Embedding |

**Detailed Experimental Settings.** The software framework includes Python 3.8.12, Pytorch 2.1.0, CUDA 12.1 and Pytorch-Geometric 2.5.3. The hardware includes Intel(R) Xeon(R) Silver 4214R CPU, 128GB RAM and NVIDIA A100 GPU. Table 6 summarizes the hyper-parameter settings of our proposed method. Here, $L$ is the number of attention blocks, $K$ is the number of attention heads, $d$ is dimension of latent features, $\delta$ is dropping rate used in Dropout, $\lambda$ is the momentum cluster-aware attention weight, $\alpha$ is the cluster-aware regularization weight, $lr$, $wd$ and $T$ are the learning rate, the weight decay and total epochs during training, respectively.

Table 6: Hyper-parameter values.

|  | $L$ | $K$ | $d$ | $\delta$ | $\lambda$ | $\alpha$ | $lr$ | $wd$ | $T$ |
|---|---|---|---|---|---|---|---|---|---|
| Cora | 2 | 4 | 128 | 0.2 | 0.2 | 0.1 | 5e-4 | 6e-6 | 1500 |
| CiteSeer | 2 | 6 | 256 | 0.2 | 0.1 | 0.2 | 5e-4 | 6e-6 | 1000 |
| PubMed | 2 | 4 | 128 | 0.2 | 0.2 | 0.2 | 5e-4 | 6e-6 | 1000 |
| Amazon-Photo | 2 | 4 | 128 | 0.2 | 0.1 | 0.1 | 5e-4 | 6e-6 | 1000 |
| Amazon-Computers | 2 | 4 | 128 | 0.2 | 0.2 | 0.2 | 5e-4 | 5e-4 | 1500 |
| Coauthor-CS | 2 | 4 | 128 | 0.3 | 0.2 | 0.1 | 9e-4 | 5e-4 | 1500 |
| Coauthor-Physics | 2 | 4 | 64 | 0.2 | 0.2 | 0.1 | 6e-4 | 5e-4 | 1500 |

## A.3 MORE VISUALIZATION ANALYSES

Figure 8 presents the visualization results of momentum attention weights between nodes for the remaining graph datasets (Cora, Amazon-Photo, Amazon-Computers, and Coauthor-CS). Compared with version $V_1$ and version $V_2$, the color of the lines outside the diagonal squares has become lighter, which means that the dependence between different clusters has been reduced. When comparing version $V_1$ with version $V_3$, significant overlaps are observed among clusters in version $V_3$, with clusters at both ends of the diagonal becoming similar and indistinguishable. This strongly demonstrates the effectiveness of our proposed modules. Compared with version $V_2$, version $V_3$ further removes momentum cluster-aware attention, so a lot of clustering information is lost and only a few easily distinguishable clusters can be solved. The visualization for the Coauthor-Physics dataset presents a more complex case due to the relatively large number of clusters. When comparing versions $V_1$, $V_2$, and $V_3$ on the diagonal, we find that the corresponding color shade satisfies $V_1 > V_2 > V_3$. This indicates that by enhancing task information, the model can enhance its focus on individual clusters. When examining the color intensity between clusters across versions $V_1$, $V_2$, and $V_3$, we find that the corresponding color shade satisfies $V_1 < V_2 < V_3$. This shows that introducing task-related constraints effectively reduces the dependence between different clusters.

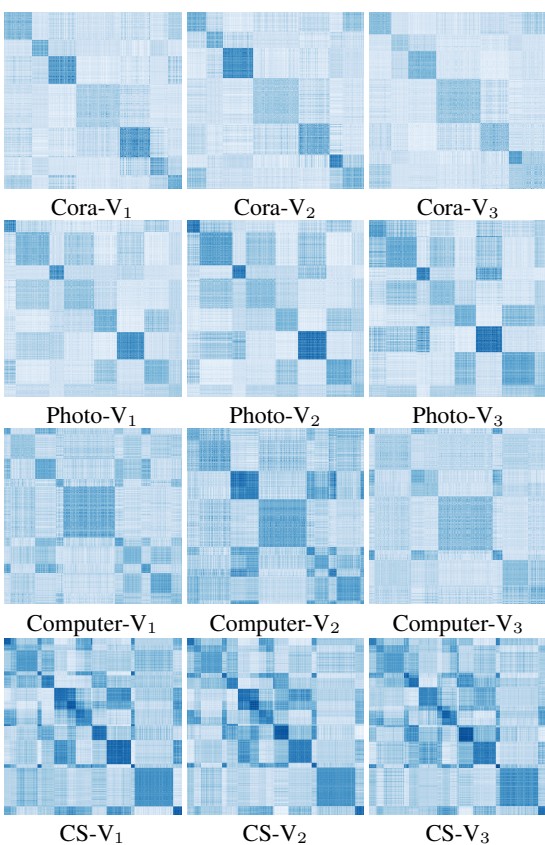

Figure 8: Attention visualization on the Cora, Amazon-Photo, Amazon-Computers and Coauthor-CS datasets. The color shade represents the different attention weight values. The darker the color, the greater the value.

