# OpenReview forum: "CTGC: Cluster-Aware Transformer for Graph Clustering"
_ICLR.cc/2025/Conference — Submitted to ICLR 2025_

### Official Review · Reviewer_5Hpk · 2024-10-28

**Soundness:** 2
**Presentation:** 3
**Contribution:** 2
**Rating:** 3
**Confidence:** 5

**Summary:**

The paper introduces an approach to graph clustering by incorporating the transformer architecture, which is particularly excel at capturing long-range dependencies. The authors propose two key mechanisms: momentum cluster-aware attention and cluster-aware regularization, to enhance the model's task awareness during representation learning.

**Strengths:**

- Alleviating the long-distance dependency problem in graph clustering is a research challenge.
- The paper is written in a very clear and accessible manner.

**Weaknesses:**

- The author claims to be the first to introduce the graph transformer, but this statement isn't entirely accurate. Several prior studies[1,2] on graph clustering have likely also incorporated graph transformers.
- How does the transformer's ability to capture long-range dependencies translate to improved clustering performance compared to traditional GNNs, and what theoretical justifications exist for this choice?
- The long-distance dependencies in graph clustering are constrained by the over-smoothing problem. Does the method proposed by the authors effectively address the over-smoothing issue in GNNs?
- Could the authors elaborate on the design choices behind the momentum cluster-aware attention mechanism, and how it contributes to the model's ability to learn more effective representations?
- Given the computational complexity of transformer models, how does the proposed model scale with larger and more complex graphs?
- The authors' method is very similar to [3]. Could you explain the differences between them?

[1] Transforming Graphs for Enhanced Attribute Clustering: An Innovative Graph Transformer-Based Method

[2] Attribute graph clustering via transformer and graph attention autoencoder

[3] Less is More: on the Over-Globalizing Problem in Graph Transformers

**Questions:**

See above.

---

> ### Author Response · Authors · 2024-11-19
> **Response to reviewer 5Hpk's comments**
>
> # W1
>
> Thank you for pointing this out. We acknowledge that there are some inaccuracies in the description with our presentation. Most current attempts to apply Transformers to graph clustering tasks, such as those in [1] and [2], involve hybrid architectures combining GNNConv/Laplacian Filters with attention mechanisms. In contrast, our method is a pure attention-based architecture, as highlighted in blue in the revised version of the pdf. Additionally, while [1] claims to be the first to apply Transformers to graph clustering, the work has not yet been officially published, meaning it has not been peer-reviewed or formally recognized in the field.
>
> Regarding the differences between our work and [1], [2], there are several key points. First, neither [1] nor [2] designed experiments to evaluate the effectiveness of the attention mechanism introduced by the transformer. As a result, it is difficult to determine whether GNNConv or attention plays a more significant role in their results based solely on performance comparisons. Secondly, the current attempts to apply Transformers to graph clustering in [1] and [2] have not yielded favorable results. Since both [1] and [2] use part of our test dataset, we have compared their results with ours, revealing a substantial performance gap. Consequently, we do not consider [1] and [2] as effective attempts to apply transformers to graph clustering.
>
> | Dataset |     Method     | ACC             | NMI             | ARI             | F1              |
> | :------: | :------------: | --------------- | --------------- | --------------- | --------------- |
> |   Cora   |     GTAGC     | 71.70           | 54.00           | 48.90           | 70.30           |
> |          |     DAEGC     | 72.60           | 52.90           | 49.60           | 71.10           |
> |          | **Ours** | **80.58** | **62.37** | **63.78** | **78.37** |
> | CiteSeer |     GTAGC     | 70.80           | 45.20           | 46.90           | 65.70           |
> |          |     DAEGC     | 69.60           | 43.00           | 45.30           | 65.30           |
> |          | **Ours** | **72.56** | **46.63** | **48.78** | **66.63** |
> |  PubMed  |     GTAGC     | 67.80           | 31.80           | 29.00           | 66.40           |
> |          |     DAEGC     | 76.30           | 37.00           | 42.10           | 75.50           |
> |          | **Ours** | **78.36** | **42.55** | **45.91** | **78.24** |
>
> # W2
>
> To further highlight the contribution of the long-range dependencies captured by our transformer model, we conducted an additional case study (which has been updated in the latest version of the pdf). Due to time constraints, the results for only CiteSeer, PubMed, and Coauthor-Physics are included at this stage. We plan to complete the results for the remaining datasets in the future. In this experiment, we set the attention scores of nodes with a shortest path distance greater than 3 to zero. As observed, the cluster divisions become less distinct compared to the original figure, with the boundaries between clusters becoming more blurred.
>
> Regarding the theoretical explanation, [3] noted that GNNs function as low-pass filters. The node representations generated by GNNs primarily capture low-frequency information, representing the common or similar features between nodes, while high-frequency information, characterized by large differences between nodes, is filtered out. The underlying assumption is that nodes are similar to their neighbors, and the process of aggregation, propagation, and stacking in GNNs serves to retain this similarity. However, to capture long-range dependencies, GNNs require stacking multiple layers. As the depth increases, excessive low-pass filtering results in the node embeddings predominantly reflecting the common features (or DC component) of the graph, leading to the over-smoothing problem where node representations become overly similar. In contrast, self-attention mechanisms in transformers allow each node to attend to all other nodes within a single layer, eliminating the need for stacking multiple layers and avoiding the over-smoothing problem. Studies [4-7] have demonstrated that transformer architectures are effective at capturing long-range dependencies in graphs and have been widely adopted for this purpose.
>
> ---
>
> [1] Transforming Graphs for Enhanced Attribute Clustering: An Innovative Graph Transformer-Based Method
>
> [2] Attribute graph clustering via transformer and graph attention autoencoder
>
> [3] Revisiting Graph Neural Networks: All We Have is Low-Pass Filters
>
> [4] Representing long-range context for graph neural networks with global attention
>
> [5] Rethinking graph transformers with spectral attention
>
> [6] Global self-attention as a replacement for graph convolution
>
> [7] Long-range transformers for dynamic spatiotemporal forecasting

---

> ### Author Response · Authors · 2024-11-19
> **Response to reviewer 5Hpk's comments**
>
> # W3
>
> In GNNs, capturing long-range dependencies typically requires stacking multiple layers, which can lead to the over-smoothing problem. As the number of layers increases, the distinctions between nodes diminish, and eventually, the node representations become overly similar at higher levels. In contrast, the transformer architecture in our method allows each node to attend to all other nodes within a single layer. Our model uses only 2layers, with most configurations having 4 attention heads. As a result, the architecture remains shallow, avoiding the over-smoothing problem.
>
> # W4
>
> The design of momentum cluster-aware attention shares similarities with the concept of cluster centers. By fusing cluster embeddings on nodes, the method brings nodes closer to their respective cluster centers while distancing them from other clusters. Common approaches for generating cluster embeddings include using the Avg, Max, or Sum of the nodes within a cluster. However, we argue that it is more effective to align this process with the method used for generating node embeddings, allowing the model to learn and optimize the parameters $Q_\mathcal{C}$ and $W_{Q_\mathcal{C}}^{(l)}, K_{Q_\mathcal{C}}^{(l)}, V_{Q_\mathcal{C}}^{(l)}$ from the data. The results of our ablation experiments, which compare cluster embeddings generated using Avg, Max, Sum, etc., further support this approach.
>
> # W5
>
> Thank you for highlighting the lack of experiments on larger graph datasets. For training transformer-based models on graphs, we can adopt the standard approach used with OGBN datasets, where batch training is performed by sampling subgraphs from the original graph. We are currently evaluating our method on larger graph datasets such as ogbn-arxiv and Reddit, and we will update the original paper with the experimental results once they are available.
>
> # W6
>
> First, the architecture of our method differs from that of [8]. Method [8] employs a hybrid architecture based on GCNConv and Attention, whereas our approach is a pure attention-based architecture. Although [8] claims that the GCN and Transformer parts independently predict node labels, the collaborative training used in their method allows the transformer to benefit from the GCN part, as demonstrated in their ablation experiments. When only the Transformer part is used, the results in [8] drop significantly, whereas our pure attention architecture achieves state-of-the-art (SOTA) performance in clustering tasks.
>
> Second, our method is specifically designed for clustering tasks, while the method in [8] focuses on node classification tasks. These tasks are fundamentally different: node classification aims to predict node labels based on the final node embeddings, whereas clustering aims to identify patterns within the graph to group nodes into distinct clusters or communities.
>
> Finally, although both our method and [8] incorporate similar fusion processing for cluster and node embeddings, the purposes and processing details differ. [8] uses METIS for graph partitioning, followed by average pooling and a linear layer fusion to obtain cluster embeddings, with the goal of capturing global information similar to a local subgraph. In contrast, we use momentum-based cluster embedding fusion, where the cluster embedding serves a role akin to a cluster center. By fusing the cluster embedding with node embeddings, we aim to bring nodes closer to their own cluster center while distancing them from other cluster centers. Additionally, the momentum update strategy we employ enhances task information, allowing the model to iteratively improve the cluster embeddings.
>
> ---
>
> [8] Less is More: on the Over-Globalizing Problem in Graph Transformers

---

> > ### Comment · Reviewer_5Hpk · 2024-11-27
> >
> > Thank you for the author's response, which has partially alleviated my concerns, and I have increased my score to 4.

---

> > > ### Author Response · Authors · 2024-11-27
> > > **Response to reviewer 5Hpk's comments**
> > >
> > > Thanks for your kind feedback and response. If you have any additional concerns regarding the acceptance of this article, please feel free to raise them, and we will address them promptly.

---

> ### Author Response · Authors · 2024-11-23
>
> Dear AC and Reviewers,
>
> $\qquad$Thank you again for the great efforts and valuable comments. We have carefully addressed the main concerns in detail. We hope you might fnd the response satisfactory. As the discussion phase is about to close, we are very much looking forward to hearing from you about any further feedback. We will be very happy to clarify any further concerns (if any).
>
> Authors

---

### Official Review · Reviewer_gUUY · 2024-10-30

**Soundness:** 2
**Presentation:** 3
**Contribution:** 2
**Rating:** 5
**Confidence:** 4

**Summary:**

This paper presents CTGC, a novel approach integrating transformers into graph clustering to effectively capture long-range dependencies. The method enhances representation learning through momentum cluster-aware attention and cluster-aware regularization, resulting in improved clustering performance on seven real-world graph datasets compared to existing SOTAs.

**Strengths:**

1.  CTGC effectively models long-range dependencies in graph clustering, overcoming the limitations of traditional GNN methods that primarily focus on local node aggregation.

2. The paper includes numerous experiments that validate the effectiveness of the proposed method, providing strong empirical support for its claims.

3. The article is well-structured, making it easy to read and follow, which enhances the understanding of the proposed methods and results.

**Weaknesses:**

The paper states, “To the best of our knowledge, we are the first to introduce a transformer for graph clustering to alleviate long-range dependency modeling.” However, existing studies have already applied transformers in graph clustering, which questions the novelty of this approach. For example[1],[2]. Including these recent works in the baseline would enhance the credibility of the claims.

[1] Li Y, Li L, Liu X, et al. Influence maximization for heterogeneous networks based on self-supervised clustered heterogeneous graph transformer[J]. Pattern Recognition, 2024, 154: 110595.

[2] Weng W, Hou F, Gong S, et al. Attribute graph clustering via transformer and graph attention autoencoder[J].

**Questions:**

What is the time complexity of the CTGC algorithm, particularly for the momentum cluster-aware attention and cluster-aware regularization mechanisms? How does it compare to traditional clustering methods?

---

> ### Author Response · Authors · 2024-11-19
> **Response to reviewer gUUY's comments**
>
> # W1
>
> Thank you for pointing this out. We acknowledge that there are some inaccuracies in the description with our presentation. Most existing attempts to apply Transformers to graph clustering tasks rely on hybrid architectures combining GNNConv and attention mechanisms. In contrast, our approach adopts a pure attention-based architecture, as highlighted in blue in the revised version of the pdf. Furthermore, prior works such as [1] and [2] did not design specific experiments to evaluate the contribution of the attention mechanism within their transformer-based frameworks. As a result, it remains unclear whether the improvements in performance stem primarily from GNNConv or the attention mechanism, as their evaluations rely solely on comparing overall performance metrics without isolating the effects of each component.
>
> To further demonstrate the contribution of long-range dependencies captured by the transformer to the results, we conducted an additional case study based on an ablation experiment (updated in the latest version of the pdf). Due to time constraints, we have included results for CiteSeer, PubMed, and Coauthor-Physics, with plans to add results for the remaining datasets in the future. In this experiment, we set the attention scores of nodes with shortest path distances greater than 3 to zero. The results show that, compared to the original figure, the clarity of cluster divisions diminishes significantly, with the boundaries between clusters becoming more blurred. This highlights the critical role of long-range dependencies in achieving effective clustering.
>
> Moreover, existing attempts to apply Transformers to graph clustering, such as [1] and [2], have shown limited success. Notably, [2] partially uses the same test datasets as our study. For clarity, we provide a comparison between their results and ours, which reveals a significant performance gap. This disparity underscores that the methods proposed in [1] and [2] cannot be considered effective implementations of transformers for graph clustering tasks.
>
> | Dataset |     Method     | ACC             | NMI             | ARI             | F1              |
> | :------: | :------------: | --------------- | --------------- | --------------- | --------------- |
> |   Cora   |     DAEGC     | 72.60           | 52.90           | 49.60           | 71.10           |
> |          | **Ours** | **80.58** | **62.37** | **63.78** | **78.37** |
> | CiteSeer |     DAEGC     | 69.60           | 43.00           | 45.30           | 65.30           |
> |          | **Ours** | **72.56** | **46.63** | **48.78** | **66.63** |
> |  PubMed  |     DAEGC     | 76.30           | 37.00           | 42.10           | 75.50           |
> |          | **Ours** | **78.36** | **42.55** | **45.91** | **78.24** |
>
> # W2
>
> As shown in our formula, momentum cluster-aware attention leverages only the query ($Q_\mathcal{C}$) associated with the selected cluster. The corresponding $W_{Q_\mathcal{C}}^{(l)}, K_{Q_\mathcal{C}}^{(l)}$ and $V_{Q_\mathcal{C}}^{(l)}$ are learnable parameters defined in $\mathbb{R}^{|\mathbb{C}|\times d}$, , where $l$ denotes the $l$-th layer. Consequently, the time complexity of this component is $O(|\mathbb{C}|^2 \times d)$. Given that $|\mathbb{C}|$ is typically small and both $|\mathbb{C}|$ and $d$ are generally constants, the computational overhead introduced by momentum cluster-aware attention is negligible compared to the original standard attention.
>
> The time complexity of the cluster-aware constraints is $O(|\mathbb{C}|^2 \times K)$, where $K$ is represents the number of attention heads. Since both $|\mathbb{C}|$ and $K$ are relatively small, the computational overhead of this component is negligible compared to standard attention. This further highlights the efficiency and superiority of our proposed cluster-aware task information supplementation method.
>
> In contrast to traditional methods, our approach is a data-driven learning method. Conventional clustering techniques, such as KMeans, Spectral Clustering, and Louvain, primarily rely on solving optimization problems or employing heuristic strategies. These methods typically leverage either the structural information of the graph or the attribute information of the nodes, which limits their performance. Furthermore, like the cluster-aware constraints we introduced, traditional methods often struggle with overlapping cluster nodes, making them less effective in more complex scenarios. Our main table presents the results for KMeans and Node2Vec, demonstrating that their performance is significantly inferior to that of learning-based methods.
>
> ---
>
> [1] Influence maximization for heterogeneous networks based on self-supervised clustered heterogeneous graph transformer
>
> [2] Attribute graph clustering via transformer and graph attention autoencoder

---

> > ### Comment · Reviewer_gUUY · 2024-11-23
> >
> > Thank you for your detailed response! However, I still have doubts about the novelty of this paper. The contributions of this work, compared to existing studies, remain unclear to me. I maintain my scores.

---

> > ### Comment · Reviewer_gUUY · 2024-11-23
> >
> > Thank you for your detailed response! However, I still have doubts about the novelty of this paper. The contributions of this work, compared to existing studies, remain unclear to me. I maintain my scores.

---

> > > ### Author Response · Authors · 2024-11-27
> > > **Response to reviewer gUUY's comments**
> > >
> > > Thanks for your kind feedback and response. If you have any additional concerns regarding the acceptance of this article, please feel free to raise them, and we will address them promptly.

---

> ### Author Response · Authors · 2024-11-23
>
> Dear AC and Reviewers,
>
> $\qquad$Thank you again for the great efforts and valuable comments. We have carefully addressed the main concerns in detail. We hope you might fnd the response satisfactory. As the discussion phase is about to close, we are very much looking forward to hearing from you about any further feedback. We will be very happy to clarify any further concerns (if any).
>
> Authors

---

### Official Review · Reviewer_vxmC · 2024-11-03

**Soundness:** 3
**Presentation:** 3
**Contribution:** 2
**Rating:** 5
**Confidence:** 4

**Summary:**

Graph clustering faces challenges with long-range dependencies and suboptimal embeddings due to a two-stage approach. CTGC addresses these by integrating transformers for better dependency modeling and introducing mechanisms to enhance task awareness, leading to improved clustering quality on real-world datasets.

**Strengths:**

1. Nice presentation with clear figures.

2. Technique sound. Lots of experiments are provided.

3. The paper is well-written and organized.

**Weaknesses:**

1. From my idea, the proposed method seems to integrate Transformer frameworks in node-level clustering, which lacks novelty. Maybe you should emphasize the importance of why you use transformer instead of other frameworks. In other words, the motivation should be more clear.

2. As for the long-range dependencies, I would recommend the authors add case studies to intuitively show that the Transformer framework indeed helps with it, rather than only relying on the performance.

3. Maybe larger datasets should be used for comparison.

4. If possible, I would like to see the performances of the proposed method on graph-level clustering.

**Questions:**

See weaknesses.

---

> ### Author Response · Authors · 2024-11-19
> **Response to reviewer vxmC's comments**
>
> # W1
>
> While transformers have been widely applied to node tasks such as classification, their application to clustering tasks remains relatively underexplored. Unlike classification, which focuses on predicting node labels using the final node embeddings, clustering aims to uncover patterns in the graph to group nodes into distinct clusters or communities.
>
> Regarding the motivation for using transformers over other frameworks like GNNs, our data analysis, as discussed in the introduction, provides compelling evidence. For example, in datasets such as Cora, CiteSeer, and PubMed, a significant proportion of shortest path distances between nodes within the same cluster exceed 3. Since most GNNs are limited to 2–3 layers, they struggle to effectively capture these long-range dependencies. While deeper GNNs can theoretically address this issue, they often suffer from challenges like over-smoothing and over-squashing, which degrade performance.
>
> Moreover, most existing attempts to apply transformers to graph clustering tasks rely on hybrid architectures that combine GNNConv and attention mechanisms. In contrast, our approach is a pure attention architecture, which not only eliminates the need to disentangle the complex contributions of GNNConv and attention mechanisms to the final result but also offers valuable insights into the design of a unified model.
>
> # W2
>
> In this study, we compared the performance of the proposed transformer framework with GNN-based methods, as presented in the main results table, to demonstrate the effectiveness of our approach. To further highlight the contribution of long-range dependencies captured by the transformer, we conducte a case study (updated in the latest version of the pdf). Due to time constraints, we have included results for CiteSeer, PubMed, and Coauthor-Physics, with plans to complete the analysis for additional datasets in future work. In the experiment, we set the attention scores for node pairs with shortest path distances greater than 3 to zero. The results show that, compared to the original graph, the cluster distinctions became significantly less pronounced, and the boundaries between clusters appeared more blurred, underscoring the importance of capturing long-range dependencies in clustering tasks.
>
> # W3
>
> Thank you for highlighting the need for experiments on larger graph datasets. We are currently evaluating our method on larger datasets, such as OGBN-Arxiv and Reddit, to further validate its scalability and effectiveness. Once the experimental results are available, we will update the original paper accordingly.
>
> # W4
>
> Clustering at the graph level and node level are fundamentally different tasks, with distinct objectives and focuses. Node clustering involves partitioning all nodes within a graph into multiple clusters, ensuring that nodes with similar attributes or structural properties are grouped closely, while those with dissimilar characteristics are positioned farther apart. Although capturing long-range dependencies is important—given that nodes in the same cluster can have considerable shortest path distances—it is generally more localized in nature. In contrast, graph-level clustering emphasizes extracting information about an entire graph from a global perspective.
>
> In our method, the momentum cluster-aware attention and cluster-aware constraints are specifically designed to optimize clustering based on intra-cluster information, rather than focusing on extracting global information from a single graph. Global graph information is inherently uniform across all nodes within a graph and lacks the variability needed for effective node-level clustering. When partitioning nodes into clusters, leveraging the localized information of distinct clusters tends to be more effective than relying solely on the global information shared across the graph.

---

> ### Author Response · Authors · 2024-11-23
>
> Dear AC and Reviewers,
>
> $\qquad$Thank you again for the great efforts and valuable comments. We have carefully addressed the main concerns in detail. We hope you might fnd the response satisfactory. As the discussion phase is about to close, we are very much looking forward to hearing from you about any further feedback. We will be very happy to clarify any further concerns (if any).
>
> Authors

---

### Author Response · Authors · 2024-11-21
**Evaluation on Large-Scale Datasets**

To comprehensively evaluate the effectiveness of our proposed method, we conducted validation on two large-scale datasets, ogbn-arxiv and ogbn-products. The experimental results, presented in Table 1, include comparative performance metrics against baseline methods cited from [1]. As shown in the table, our method consistently outperforms existing GNN-based approaches, even when applied to larger datasets, thereby demonstrating its scalability and superior efficacy.

Table 1: Clustering performance of CTGC and five baselines on larger graph datasets. The bold and underlined scores indicate the best and second best results respectively.

|    Dataset    | Metric | KMeans | Node2Vec |       S$^3$GC       | DMoN  |         MAGI         |      Ours      |
| :-----------: | :----: | :----: | :------: | :-------------------: | ----- | :-------------------: | :-------------: |
|  ogbn-arxiv  |  ACC  | 18.10 |  29.00  |         35.00         | 25.00 | $\underline{38.80}$ | **41.09** |
|              |  NMI  | 22.10 |  40.60  |         46.30         | 35.60 | $\underline{46.90}$ | **47.58** |
|              |  ARI  |  7.40  |  19.00  |         27.00         | 12.70 | $\underline{31.00}$ | **34.36** |
|              |   F1   | 12.90 |  22.00  |         23.00         | 19.00 | $\underline{26.60}$ | **28.52** |
| ogbn-products |  ACC  | 20.00 |  35.70  |         40.20         | 30.40 | $\underline{42.50}$ | **43.20** |
|              |  NMI  | 27.30 |  48.90  |         53.60         | 42.80 | $\underline{55.10}$ | **55.96** |
|              |  ARI  |  8.20  |  17.00  | $\underline{23.00}$ | 13.90 |         21.50         | **24.80** |
|              |   F1   | 12.40 |  24.70  |         25.00         | 21.00 | $\underline{27.60}$ | **28.52** |

---

[1] Revisiting Modularity Maximization for Graph Clustering: A Contrastive Learning Perspective

---

### Meta-Review · Area_Chair_Vw4t · 2024-12-16

**Metareview:**

This paper focuses on two issues in graph clustering: long-range dependencies capturing difficulty and two-stage clustering. It utilizes transformer as crucial graph encoder, and designs momentum cluster-aware attention and cluster-aware regularization to enhance the encoder task-awareness. In momentum cluster-aware attention, the clustering results are utilized to guide the node embedding production via cluster-aware queries. In cluster-aware regularization, the cluster information is fused into bordering nodes by minimizing the overlap between clusters while maximizing the cluster completeness.   Experimental results validate the effectiveness of proposed scheme.


After rebuttal, all reviewers are still negative about this paper. Especially, the importance of why utilizing transformer rather than other frameworks is not clear. Existing studies have already applied transformers in graph clustering, and the proposed method is very similar to existing works. So, the novelty of proposed approach is limited. The mechanism of momentum cluster-aware attention is not elaborated.
The authors are encouraged to further perfect this manuscript according to relevant comments.

**Additional Comments On Reviewer Discussion:**

All reviewers are still negative about this paper. Especially, the importance of why utilizing transformer rather than other frameworks is not clear. Existing studies have already applied transformers in graph clustering, and the proposed method is very similar to existing works. So, the novelty of proposed approach is limited. The mechanism of momentum cluster-aware attention is also not elaborated.

---

### Decision · Program_Chairs · 2025-01-22

Reject